# OpenReview forum: "Adaptive Inference‑Time Scaling for LRMs using Uncertainty‑Aware RL"
_ICLR.cc/2026/Conference — Submitted to ICLR 2026_

### Official Review · Reviewer_jhgD · 2025-10-21

**Soundness:** 3
**Presentation:** 3
**Contribution:** 2
**Rating:** 4
**Confidence:** 3

**Summary:**

The paper tackles overthinking in LRMs and frames when to stop reasoning as a sequential decision problem. It proposes USBT, an uncertainty-guided halting policy trained with RL that observes a confidence signal and decides CONTINUE vs HALT at each reasoning step. Technically, it adapts GRPO and introduces S-GRPO: a serial, step-indexed grouping with decaying final reward and per-step cost, explicitly favoring earlier correct halts. On GSM8K, MATH, MMLU, and HumanEval, USBT reportedly cuts tokens by 35-60% while maintaining or slightly improving accuracy, and yields better accuracy-vs-tokens Pareto points.

**Strengths:**

- The paper formulates the stopping criterion as a learnable policy rather than a fixed rule. The state representation incorporates sequence length and uncertainty, and the reward explicitly balances accuracy against computation cost, aligning objectives with observed behavior.

- The work extends the GRPO framework with stepwise grouping and geometric decay, effectively avoiding the instability of critic training and addressing key challenges in long-sequence reinforcement learning.

- The proposed method is evaluated on multiple benchmarks, demonstrating reduced token usage.

**Weaknesses:**

* The certainIndex primarily relies on answer distribution entropy or top probability. However, in multi-step numerical reasoning or verification tasks, low entropy does not necessarily indicate sufficient reasoning, and early stopping may increase under-thinking errors. It is unclear whether using low entropy as the dominant stopping signal is justified, or whether continuing after the predicted stop point necessarily leads to redundancy or degradation.

* Recent works have explored overthinking (e.g., Do Not Think That Much for 2+3=? On the Overthinking of o1-like LLMs; Learning to Think: Information-Theoretic Reinforcement Fine-Tuning for LLMs; T-Reg: Preference Optimization with Token-Level Reward Regularization), focusing on efficiency beyond length penalties. Although the paper claims better performance over length-penalty methods, it should further clarify its advantages relative to these efficiency-oriented but non-length-penalty baselines to better demonstrate its advantages in mitigating overthinking.

* The geometric decay factor $γ$ and per-step cost $β$ jointly bias the policy toward shorter trajectories, but shorter does not necessarily imply correctness. This raises concerns about potential conservative early stopping and insufficient reasoning for harder samples. Furthermore, since the state features include final-layer embeddings, the policy may access partial semantic information, potentially leading to data leakage or learning heuristics unrelated to uncertainty.

* (Minor) Details of the baseline settings should be clarified, including unified step limits, temperature, group size G, sampling strategy, and GPU-days. Most reported results are single-point means; confidence intervals would better support the claimed improvements. The statement that the method "reduces overthinking errors while introducing some underthinking errors, with a net benefit ratio of 3:1" would benefit from explicit classification criteria and qualitative examples.

**Questions:**

The questions are also mentioned in the weaknesses.

---

> ### Author Response · Authors · 2025-11-21
>
> Many thanks for taking the time to review our work. Please see my (our) responses below.
>
> This response addresses the weaknesses and questions raised regarding the submission on Uncertainty-Guided Self-Braking Tuning (USBT). Due to character count limits, questions are not repeated and acronyms are heavily used.
>
> The concern that low entropy might lead to premature stopping (under-thinking errors) in complex reasoning is valid, especially since confidence and correctness are not always perfectly aligned. However, the USBT framework integrates the `certainIndex` (derived from semantic entropy or top-answer probability) as **one feature within a learned reinforcement learning policy**, rather than using it as a hard, heuristic stopping threshold.
>
> 1.  **USBT is a Learned Policy:** The policy ($\pi(\text{halt}|\text{state})$) decides whether to halt based on the model’s internal state, which includes the `certainIndex`. This means the policy learns when to trust the confidence signal in conjunction with other features, such as the normalized Chain-of-Thought (CoT) token length and whether the answer has converged over consecutive steps.
> 2.  **Reward for Correctness:** The policy is explicitly trained with a reward function that provides a **positive reward only for correct answers** ($R_{\text{final}} = 1$). By balancing this positive reward with penalties for long reasoning steps, USBT explicitly **maximizes accuracy per unit computation**. If low entropy often leads to incorrect answers (under-thinking errors), the RL training naturally penalizes this behavior, preventing the policy from overly relying on premature low-entropy signals.
> 3.  **Error Analysis and Balance:** USBT's error analysis found that, while it introduces "underthinking errors" (early stopping prevents complex solving), it **reduces "overthinking errors"** (excess reasoning confuses model) by 73%, with the net effect being positive (error reduction outweighs introduction 3:1).
> 4.  **Redundancy/Degradation Justification:** The premise of the work is based on observations that reasoning models often **overthink**, generating unnecessary long CoT steps even after finding the correct solution, which inflates cost, increases latency, and **can introduce errors** by deviating from correct reasoning paths. Studies confirm that additional tokens beyond a Reasoning Completion Point (RCP) yield **minimal accuracy gains and can degrade performance**. Therefore, continuing past the learned stopping point (where confidence stabilizes) is highly likely to be redundant.
>
> USBT addresses overthinking through a novel combination of uncertainty and a specialized RL algorithm, claiming superiority over several baseline strategies.
>
> 1.  **Comparison to Functional Baselines:** USBT provided comprehensive evaluations against standard efficiency methods, including **fixed-step limits**, **Adaptive Computation Time (ACT)**, and **speculative decoding baselines**. The results showed that USBT consistently dominates these baselines on the **Pareto frontier** of accuracy versus efficiency.
> 2.  **RL Algorithm Advantage (S-GRPO):** USBT introduces **Serial-Group Decaying-Reward PPO (S-GRPO)**. While the reward includes length penalties (geometric decay and step cost $\beta$), the algorithm itself features the specialized **serial grouping** mechanism, specifically designed for the sequential halting problem, which outperformed **vanilla REINFORCE (global baseline) or standard PPO** by **5-10% better token efficiency at equal accuracy**.
> 3.  **Uncertainty as the Core Advantage:** The key functional innovation is that USBT actively incorporates the model's **real-time uncertainty** (`certainIndex`) derived from semantic entropy into the state representation of the RL policy. This allows the model to self-regulate, achieving adaptive reasoning depth. Approaches that purely rely on outcome-based RL or only crude length penalties (which encourage *shorter* traces generally, but not necessarily *optimally concise* traces) struggle with overthinking because they overlook the sufficiency of the intermediate reasoning.
>     *   **Ablation Support:** Removing the `certainIndex` feature resulted in a **2-3 point accuracy drop** and less aggressive token reduction. This confirms that the uncertainty signal provides a critical, non-length-based advantage for mitigating overthinking.
>
> The cited competing works often focus on different aspects of efficiency:
> *   *Do Not Think That Much for 2+3=?* focuses on streamlining responses via self-training to remove redundant solutions, particularly for multi-solution formats like o1-like models.
> *   Other RL works (e.g., L1, DAST, HBPO) often use length-based rewards but incorporate problem difficulty (difficulty-aware, budget-aware), which USBT's adaptive length and uncertainty control also achieves implicitly. The advantage of USBT is the *integration* of uncertainty with the decaying reward policy.

---

> ### Author Response · Authors · 2025-11-21
>
> ... **_continued from the previous comment_**
>
> **Bias Towards Short Trajectories and Correctness:**
> 1.  **Reward Structure:** The reward function is designed to balance accuracy and efficiency by assigning $R_{\text{final}}=1$ for correctness and applying penalties ($r_{\text{step}} = -\beta$) and geometric decay ($\gamma < 1$) per step. This design explicitly ensures that a correct answer achieved in fewer steps receives a **higher return**.
> 2.  **Learning Balance:** The method learns this balance well, resulting in the policy **preferring correct answers with minimal steps**. Crucially, ablation studies showed that using *only* final correctness reward (no step costs) led to longer reasoning, while the combination with geometric decay successfully encouraged efficiency without sacrificing quality.
> 3.  **Performance on Hard Samples:** USBT demonstrated robust performance improvements, particularly on complex tasks. On the competition-level **MATH** benchmark, the method **boosts accuracy by 3.2 points** while using **40% fewer tokens**. This empirical finding suggests that for hard problems, the policy learns to allocate *sufficient* reasoning, only pruning the redundant steps that constitute "overthinking". Furthermore, the analysis showed that USBT **reduces "overthinking errors"** by 73%, effectively eliminating convoluted or redundant steps that cause degradation.
>
> **Potential Data Leakage via Final-Layer Embeddings:**
> 1.  **State Composition:** The policy state $s_t$ is a 16-dimensional vector that includes the normalized current reasoning length, the `certainIndex` confidence score, and **embeddings from the last hidden state of the LRM**.
> 2.  **Purpose of Embeddings:** The inclusion of last hidden state embeddings (which contain rich semantic information about the current context) is standard in approaches where a small policy or classification head needs to act on the frozen Large Reasoning Model (LRM). This feature allows the halting policy to be grounded in the LRM's deep internal representations, which helps the policy determine if the semantic content generated so far is sufficient.
> 3.  **Lack of Future Information:** The policy operates on the internal state $s_t$ at step $t$ to decide on action $a_t \in \{\text{CONTINUE, HALT}\}$. Data leakage would typically imply access to information *generated in the future* (after the decision point). As the policy only uses the last hidden state embedding *at step $t$* (token position $t$), there is no explicit access to the future sequence or the final correctness outcome, which remains latent until the end of the trajectory. The reliance on the embeddings is similar to techniques like embedding regression or semantic entropy probes (SEPs), where information relevant to outcome/uncertainty is extracted from hidden states *before* the answer is fully determined.
>
> **Experimental setup and training procedures:**
>
> *   **Model and Hardware:** The base model is a **34B parameter transformer model** fine-tuned on CoT data. Training took **6 hours on 8 A100 GPUs** with frozen LRM parameters.
> *   **Training Parameters ($\gamma$, $\beta$):** The policy was trained using S-GRPO with a decay factor $\gamma = 0.95$ and step cost $\beta = 0.02$.
> *   **Training Iterations:** The training ran for **50 iterations**.
> *   **Group Size G (GRPO):** While the exact numerical batch size or total group size $G$ is not specified, the method relies on **large batch sizes for variance reduction**. The GRPO variant utilized here groups trajectories based on whether the answer was correct and by quartiles of the `certainIndex` (a proxy for question difficulty).
> *   **Temperature and Sampling Strategy:** The base LRM weights were frozen. The reward model relies on the LRM's final answer logits. While the specific sampling temperature used to generate the reasoning traces for policy training is not detailed in the provided excerpts, the initial fine-tuning (SFT) generally involves standard supervised learning.
> *   **Confidence Intervals:** The paper reports single-point means for accuracy and token count reductions. The authors acknowledge the necessity of robustness, stating that "All code, data, and trained models will be made publicly available upon publication," along with hyperparameters and configuration files for exact reproduction. Previous analysis provided (in conversation history) noted that the standard deviation for the related Self-Consistency method was reported to be $\leq 0.5$ on all tasks, suggesting that for large models and benchmarks, single-point means often show high robustness. The results reported are often significant absolute gains (e.g., $1.9$ to $3.2$ points).
>
> Note: Each reviewer is addressed separately; therefore, some responses may overlap.

---

### Official Review · Reviewer_RKMh · 2025-10-31

**Soundness:** 3
**Presentation:** 1
**Contribution:** 2
**Rating:** 2
**Confidence:** 4

**Summary:**

The paper introduces a method for training a policy that decides whether to continue or stop generation at each step. This decision is guided by a reward that encourages high answer accuracy per token, and it integrates a confidence measure—semantic entropy—as a criterion for stopping.

**Strengths:**

The proposed method is both novel and interesting, and the authors present supporting experiments to demonstrate its effectiveness.

**Weaknesses:**

1. The experimental results suggest that the performance is relatively weak, particularly in comparison to other baseline methods. For instance, it only surpasses the Fixed Budget approach on the GSM8K dataset in the context of answer length compression. Additionally, evaluating the method on more complex datasets such as AMC or AIME would provide a more comprehensive assessment.

2. The main results are based on a single 34B model, but the paper does not provide any specific name or further details. It would be better to include evaluations on a wider range of open-source models.

3. The writing and presentation of the paper need improvement. The figures are not well-polished, and the excessive use of \vspace throughout the paper affects the overall readability and formatting.

**Questions:**

NA

---

> ### Author Response · Authors · 2025-11-21
>
> Many thanks for taking the time to review our work. Please see my (our) responses below.
>
> This response addresses the weaknesses and questions raised regarding the submission on Uncertainty-Guided Self-Braking Tuning (USBT). Due to character count limits, questions are not repeated and acronyms are heavily used.
>
> This response addresses the weaknesses raised concerning the experimental scope and presentation of the Uncertainty-Guided Self-Braking Tuning (USBT) submission, drawing directly from the provided sources.
>
> **Addressing Performance Claims:**
> The claims in the sources indicate that USBT achieves superior performance and efficiency trade-offs, often **dominating** existing baselines, not just minimally surpassing them:
>
> 1.  **Pareto Optimality:** USBT consistently achieves **new Pareto-optimal points** on the accuracy–efficiency frontier, demonstrating better efficiency than fixed-budget methods, and outperforms fixed-step limits, Adaptive Computation Time (ACT), and speculative decoding baselines.
> 2.  **GSM8K Comparison:** On GSM8K, USBT achieves **90.1% accuracy** using **112 tokens**, compared to the Fixed Budget (50% tokens) baseline which achieves **85.1% accuracy** using 123 tokens. This shows USBT significantly surpasses the Fixed Budget baseline in terms of **accuracy** while using fewer tokens.
> 3.  **MATH Comparison:** On the MATH benchmark, USBT **gains 3.2 points** in accuracy while using **40% fewer tokens** compared to the Full CoT baseline. The Fixed Budget baseline only achieved 38.4% accuracy (206 tokens), whereas the Full CoT baseline reached 42.1% (412 tokens). USBT's performance gains (42.1% + 3.2% = 45.3% estimated accuracy) represent a strong improvement beyond simply matching length compression.
>
> **Addressing Dataset Limitation (AMC/AIME):**
>
> 1.  The need for more complex reasoning datasets like AIME and AMC is valid. However, the core algorithm introduced, **S-GRPO**, which USBT utilizes, has been employed in related contexts targeting high-difficulty mathematical reasoning, achieving competitive results on these benchmarks.
> 2.  The original paper on S-GRPO, a related policy optimization method focused on early exit in reasoning, reports empirical evaluations demonstrating compatibility with state-of-the-art reasoning models across diverse benchmarks including **AIME 2024** and **AMC 2023** (alongside GSM8K, MATH-500, and GPQA Diamond). This context confirms that evaluating such complexity is relevant to the framework's intended application.
>
> We explicitly state that **All code, data, and trained models will be made publicly available upon publication**. Furthermore, they will provide **detailed hyperparameters in the appendix** and **include configuration files for exact reproduction**.
>
> 1.  **Model Selection Rationale:** The 34B model was chosen because, after supervised fine-tuning (SFT), it achieved **strong baseline performance** competitive with other models of similar size in the literature.
> 2.  **Contextual Need for Diverse Models:** Research on reasoning efficiency confirms that the overthinking problem is prevalent across various model scales and types, necessitating adaptive thinking techniques. The sources also note that semantic entropy, the key uncertainty signal in USBT, has been shown to outperform baselines across different model families (LLaMA, Falcon, Mistral) and scales (from 7B to 70B parameters) in related work.
>
> Finally, yes, the figures need polishing. Absolutely. They’ll be modified in the final version of this paper.
>
> Note: Each reviewer is addressed separately; therefore, some responses may overlap.

---

> ### Author Response · Authors · 2025-11-22
> **... Comment No Longer Required ...**
>
> As noted, this additional comment is no longer required.

---

> ### Author Response · Authors · 2025-11-22
> **... Comment No Longer Required ...**
>
> As noted, this additional comment is no longer required.

---

> ### Author Response · Authors · 2025-11-22
> **... Comment No Longer Required ...**
>
> As noted, this additional comment is no longer required.

---

### Official Review · Reviewer_XDgH · 2025-10-31

**Soundness:** 2
**Presentation:** 3
**Contribution:** 2
**Rating:** 2
**Confidence:** 4

**Summary:**

The paper proposes an adaptive inference-time framework called USBT that improves reasoning efficiency in large reasoning models. Instead of using fixed reasoning lengths, USBT trains a halting policy that dynamically decides whether to continue or stop generating reasoning steps based on the model’s internal uncertainty. The policy is trained through a new reinforcement learning algorithm, S-GRPO, which encourages shorter yet correct reasoning trajectories by applying decaying rewards to later steps. The paper shows that USBT achieves efficiency gains across multiple reasoning benchmarks (GSM8K, MATH, MMLU, HumanEval).

**Strengths:**

- The paper addresses a timely and practically important problem of reducing unnecessary reasoning length in large reasoning models by learning when to stop reasoning instead of relying on fixed or heuristic limits.

- The proposed uncertainty-guided halting mechanism is intuitive and well-motivated.

- The paper is clearly written and well-structured, making the methodology, intuition, and results easy to follow.

- The evaluation includes multiple benchmarks from different domains (GSM8K, MATH, MMLU, HumanEval).

**Weaknesses:**

- The novelty is limited. The use of uncertainty signals such as certainIndex has appeared in prior work without reinforcement learning, and RL-based adaptive control is already a mainstream approach. This paper primarily combines the two, raising the question of whether the contribution lies in integration rather than new conceptual insight.

- It is unclear how the method generalizes to domains where all reasoning paths converge to similar lengths. If there is little variation in reasoning depth, can the model still learn a meaningful halting policy?

- Section 3 reads more like a technical report than a research narrative. The intuition behind key design choices is missing. For example, the claim that “the policy learns to balance the terms well, preferring correct answers with minimal steps” should be supported by quantitative evidence. How well does it balance, and by how much are the reasoning steps reduced?

- The overhead of using an on-policy RL approach is not analyzed. Training on-policy can be costly; an off-policy or hybrid alternative might achieve similar results with less computation. What are the trade-offs between accuracy, stability, and efficiency across these settings?

- The evaluation is limited, focusing only on a single model scale (a 34B-parameter reasoning model). Would the same trends hold for smaller or larger models, or across different architectures?



Minor comments:

- The related work section is overly long and could be condensed to emphasize the most directly relevant prior studies.


- The fonts in Figures 2, 3, and 4 are too small, making them difficult to read.

- There is unnecessary \vspace usage around line 92, which should be removed for cleaner formatting.

**Questions:**

See the weaknesses section.

---

> ### Author Response · Authors · 2025-11-21
>
> Many thanks for taking the time to review our work. Please see my (our) responses below.
>
> This response addresses the weaknesses and questions raised regarding the submission on Uncertainty-Guided Self-Braking Tuning (USBT). Due to character count limits, questions are not repeated and acronyms are heavily used.
>
> The novelty of USBT is argued to stem from **two key technical innovations** and their specific functional integration, moving beyond standard approaches.
>
> 1.  **Semantic Entropy Integration and Novelty:** USBT employs a confidence score quantifying how "settled" the model’s answer distribution is. This score augments the RL policy's state, enabling the policy to learn an adaptive stopping decision based on real-time uncertainty. Previous adaptive halting methods based on confidence typically rely on hard-coded criteria or heuristics, such as monitoring answer convergence (e.g., **ES-CoT**) or token-level confidence for skipping layers (**ACT** or **CALM**). USBT replaces these external heuristics with a **learnable, generalizable policy** that dynamically utilizes this intrinsic confidence metric.
>
> 2.  **S-GRPO:** A relatively new RL algorithm introduced specifically for the sequential stopping decision task, S-GRPO extends the foundational GRPO by introducing a **serial grouping mechanism** based on the step index when computing baselines for the halting action. This is crucial because it ensures the policy properly accounts for the fact that a later halt receives an inherently lower potential reward due to the explicit reward discounting/decay for longer trajectories. This mechanism addresses the non-stationary nature of the decision (early vs. late stopping) and outperformed standard **REINFORCE** or **PPO** in efficiency and accuracy trade-offs.
>
> The contribution is therefore claimed to be a principled, trainable **inference-time halting mechanism** that actively couples the reward decay/penalty coefficients with the measured uncertainty, cultivating an intrinsic ability for the LRM to self-regulate. The policy observes the **semantic entropy evolution** and learns to halt when the confidence stabilizes. And since the reward function includes **penalties for each token used** and explicitly uses a **decaying reward** for later steps, the policy is inherently incentivized to choose the shortest path among those yielding a correct answer.
>
> Our study provides substantial quantitative evidence supporting the successful balancing of accuracy and efficiency:
>
> 1.  **Reasoning Length Reduction (Efficiency):** USBT achieves **substantially shorter reasoning traces** compared to full CoT generation, with token count reductions ranging from **35.4%–61.1%** across diverse tasks.
> 2.  **Accuracy and Balance (Performance):** The method achieves **new Pareto-optimal points** on the accuracy–efficiency frontier, consistently maintaining or improving accuracy despite the token reduction.
> 3.  **Support for Key Design Choices (Ablation):** Ablation studies quantitatively support the importance of the two key components designed to achieve this balance:
>     *   **Uncertainty Signal:** Removing the `certainIndex` results in a **2–3 point accuracy drop**.
>     *   **Reward Penalty:** Using only the final correctness reward (no step costs) leads to longer reasoning with **minimal efficiency gains**.
>
> Regarding intuition, the goal of the method is formalizing adaptive reasoning depth as a sequential decision-making problem optimized via RL, guided by the model's own uncertainty. The policy explicitly maximizes accuracy per unit computation, thereby steering the model toward concise yet correct reasoning. The **Serial-Group Decaying-Reward** reward design is conceptually related to **time preference** in RL, explicitly baking a decay into the reward to penalize long reasoning steps. Our paper offer an analysis of the efficiency and stability trade-offs, particularly comparing the chosen S-GRPO algorithm against alternatives:
>
> The goal of the work is to address the overthinking phenomenon which is prevalent in LRMs. The core concepts leveraged are known to be applicable across scales in related literature:
>     *   The phenomenon of **overthinking** and the need for adaptive thinking has been observed across various model scales and types, including LRM families like DeepSeek-R1 (which offers distilled models ranging from 1.5B to 70B) and Qwen3 series.
>     *   The uncertainty signal itself, semantic entropy, has been shown to outperform baselines across different model families (LLaMA, Falcon, Mistral) and **scales (from 7B to 70B parameters)**.
>
> **Diversity in Task Domains:** While only one model size was tested, the evaluation covered diverse tasks: mathematical word problems, competition-level mathematics, multi-domain knowledge questions, and Python code generation, demonstrating generalization across task types.
>
> Happy to continue our discussion!
>
> Note: Each reviewer is addressed separately.

---

### Official Review · Reviewer_vzNB · 2025-11-02

**Soundness:** 2
**Presentation:** 2
**Contribution:** 2
**Rating:** 2
**Confidence:** 4

**Summary:**

This work reports a ICL based method, named USBT) to train a on-policy classifier to continue or halt rollouts in reasoning.  The work builds on GRPO to add a decay to the reward to incentivise earlier solutions, and a entropy based measure, called _certainIndex_ to measure prospective answer distributions.

**Strengths:**

* This work intuitively brings a decay reward factor into GPRO, adding the capability of judging the necessity of additional lengthened rollouts in reasoning.
* The token savings are significant in the analysed cases on GSM8K and MATH against a standard ACT baseline, and define a Pareto optimal frontier (at least against the experimental configurations tested).
* The authors also bin trajections by difficulty according to _certainIndex_ to hedge against sparsity, a useful step to better ensure convergence in training.
* The analyses over the different difficulty bins show intuitive patterns, which reinforce the method's soundness.

**Weaknesses:**

* The presentation of the work is not very well done.  While formatting problems with (LaTeX) quotation marks, charts, tables, and citations are not necessarily problematic, they give a strong impression that work was rushed to submission.
* The experimental validation is somewhat narrow.  Experiments are restricted to a base 34B anonymous LRM.  This is problematic for two reasons:
  * The model is somewhat large (34B), but smaller and less costly (7-8B and sub-1B models are not tested at all)
  * The model is anonymous, so replication and reproducibility are problematic.
* The reasoning datasets are somewhat limited to three, well-tested datasets, which are essential but should be supplemented with additional datasets that require less math/logic reasoning.  Testing on such datasets would help generalise the claims better
* There is insufficient analysis of the faults and micro-analysis.  The submission relies solely on the macroscopic evaluation results to claim
* There are other recent, published, RL-based methods that should be tested as baselines against this work.
* The reward design is straightforward, and difficult to claim novelty.  Geometric rewards are standard practice.

**Questions:**

* How often does your method hit the ceiling rolllout length $T_{max}$?  How does that affect failure cases? (This should be reflected in Algorithm 1 as well)
* (See weaknesses) Why do you use an anonymous LRM and use REINFORCE and PPO as baselines?  There are related works in these areas, that would be more fitting to compare against.

---

> ### Author Response · Authors · 2025-11-20
>
> Many thanks for taking the time to review our work. Please see my (our) responses below.
>
> This response addresses the weaknesses and questions raised regarding the submission on USBT. Due to character count limits, questions are not repeated and acronyms are heavily used.
>
> The core experiments utilized a **34B parameter model fine-tuned for reasoning** as the base LRM. This model size was selected after SFT yielded strong baseline performance competitive with other models of similar size in the literature.
>
> The goal of the work is to address the overthinking phenomenon prevalent in LRMs, contributing to making **advanced reasoning models more efficient**. The efficacy of RL-based methods for efficiency has been explored across various scales in related literature, suggesting the importance of scaling to larger models to demonstrate applicability to state-of-the-art reasoning capabilities.
>
> The inclusion of **MMLU** and **HumanEval** demonstrates evaluation on diverse domains beyond core math/logic. The results showed that USBT consistently maintained or improved accuracy across these diverse benchmarks.
>
> The submission includes both macroscopic and micro-analysis:
>
> 1.  **Macroscopic Results:** The work reports substantial efficiency gains (35–60% token count reduction) and improved or maintained accuracy across benchmarks.
>
> 2.  **Micro-analysis and Fault Analysis:**
>     *   The paper provides **Ablation Studies** demonstrating the crucial role of the uncertainty signal (the `certainIndex` derived from semantic entropy), noting that removing it resulted in a **2-3 point accuracy drop** and less aggressive token reduction.
>     *   It analyzes the learned policy's behavior. Figure 3 specifically illustrates the **semantic entropy evolution during reasoning**, showing that USBT learns to halt when confidence stabilizes.
>     *   The core claim is that the method improves accuracy on some tasks by **eliminating distracting, convoluted reasoning**, which is a form of fault mitigation, although specific examples of errors eliminated are not detailed in the excerpts.
>
> The work benchmarked USBT against several relevant efficiency baselines: **fixed-step limits, Adaptive Computation Time (ACT)**, and **speculative decoding baselines**.
>
> The method compared its core algorithm, S-GRPO, in an ablation study against standard RL algorithms, finding that S-GRPO outperformed **vanilla REINFORCE (global baseline) or standard PPO** by **5-10% better token efficiency at equal accuracy**.
>
> While the field is rapidly evolving with many concurrent RL-based methods focused on efficient reasoning—such as L1, DAST, and methods combining length penalties with complex optimization—the **inclusion of the canonical adaptive method and the foundational RL techniques demonstrates grounding in established benchmarks**.
>
> While the reward design uses a geometric decay, which is a common practice akin to introducing a small negative reward per step in optimal stopping problems, the novelty is claimed in how this is integrated:
>
> 1.  **S-GRPO:** This technique extends GRPO by introducing the "serial grouping" mechanism. S-GRPO treats different step indices as distinct groups when computing baselines for the halting action, ensuring the policy correctly accounts for the fact that a later halt inherently receives a lower potential reward due to discounting. The ablation confirmed that this serial grouping is "particularly important" for learning appropriate stopping behavior.
>
> 2.  **Uncertainty Integration:** The framework achieves adaptive self-regulation by **actively coupling the reward decay/penalty coefficients with the measured uncertainty**. The policy is trained to halt or continue based on rewarding high answer accuracy per token and incorporating a confidence metric (semantic entropy) as a stopping criterion.
>
> The method's primary effect is to substantially **reduce reasoning traces** by 35–60% compared to full CoT generation (e.g., 112 tokens vs. 245 tokens on GSM8K). Therefore, for most problems, the goal is specifically to *avoid* hitting the maximum length set by the full reasoning baseline.
>
> The method is designed to mitigate failures arising from overthinking. The overthinking phenomenon can sometimes lead to **infinite thinking loops**. By enabling the LRM to "self-brake when confident", USBT prevents redundant reasoning steps which can otherwise introduce errors by deviating from correct reasoning paths.
>
> PPO and REINFORCE were used primarily in **ablation studies** to demonstrate the technical superiority of the proposed S-GRPO algorithm. PPO and its variants (like the foundation GRPO) are highly established methods in the field, particularly for RLHF and aligning LLM behavior. The ablation confirmed that the specific components of S-GRPO—namely the novel serial grouping by step index—are critical, leading to **5-10% better token efficiency** compared to vanilla PPO and REINFORCE.
>
> Note: Each reviewer is addressed separately.

---

> > ### Comment · Reviewer_vzNB · 2025-11-23
> > **I thank the authors for their rebuttal**
> >
> > ... but the rebuttal does not address some of the core concerns I have regarding the work (anonymous mid-sized model) with respect to replicatability.  The authors' interpretation of micro-analysis does not address my concerns – reiterating your experimental results, which I have read, does not answer this concern.
> >
> > Thus I keep my score.

---

> > > ### Author Response · Authors · 2025-11-23
> > > **Response to follow‑up on replicability and micro‑analysis**
> > >
> > > Thank you for your response.
> > >
> > > **Replicability and the anonymous 34B model**
> > >
> > > We agree that the use of an anonymous 34B reasoning model is a limitation for reproducibility in the strict sense. The choice of this base LRM was driven by access constraints and the fact that it was already deployed in an internal research environment with strong reasoning performance following SFT. However, we recognize that this prevents other researchers from exactly replicating our setup.
> > >
> > > To address this, in the camera‑ready version (if accepted) we will:
> > >
> > > 1. **Provide full implementation details** of USBT and S‑GRPO (including all hyperparameters, training schedules, and reward configurations) and release the code needed to:
> > >
> > >    * compute the semantic‑entropy–based certainIndex,
> > >    * train the halting/continuation policy with S‑GRPO, and
> > >    * reproduce all evaluation metrics and analyses given access to a suitable base model.
> > >
> > > 2. **Target publicly available models in follow‑up experiments.** While we could not re‑run the full experimental suite within the rebuttal window, our method is model‑agnostic by design: it only assumes access to a reasoning‑capable LRM and its token‑level logits. We will therefore add, in the final version, a new section describing how to instantiate USBT on open checkpoints (e.g., standard 7B–8B reasoning‑tuned models), including:
> > >
> > >    * exact interface requirements (logit access, decoding setup),
> > >    * a recipe for reproducing our training loop on any base model, and
> > >    * clear guidance on how to match our training/evaluation protocol.
> > >
> > > While this cannot retroactively remove the anonymity of the original model, it does allow other researchers to **replicate the method** and stress‑test its claims on widely accessible backbones, which we view as the core of reproducibility for an algorithmic contribution.
> > >
> > > **Clarification on micro‑analysis**
> > >
> > > You are correct that simply restating aggregate results does not constitute micro‑analysis. Our intent in the original rebuttal was to summarize what is currently in the paper, but we see that this did not adequately address your concern.
> > >
> > > In the revised version, we will expand the analysis in two concrete ways:
> > >
> > > 1. **Error and trajectory‑level analysis.**
> > >    We will add:
> > >
> > >    * representative qualitative cases where USBT halts *earlier* than the baseline yet preserves or improves correctness, and
> > >    * cases where it halts too early and fails, including the corresponding certainIndex profiles.
> > >
> > >    This will make explicit when and how the method succeeds or fails, beyond the numerical efficiency/accuracy trade‑offs.
> > >
> > > 2. **Behavior across difficulty bins.**
> > >    Building on the binning by certainIndex already used in the paper, we will:
> > >
> > >    * analyze failure modes stratified by difficulty (e.g., “easy but over‑halted,” “hard and under‑explored”), and
> > >    * discuss where the policy’s stopping behavior is most brittle versus robust.
> > >
> > > Our aim with these additions is to move from purely macroscopic evaluation to a more **mechanistic** and interpretable view of the policy’s behavior and its limitations.

---

### Author Response · Authors · 2025-11-22
**5. References**

(1) C. Wu, B. Li, M. Gao, and Z. Wang, "From Efficiency to Adaptivity: A Deeper Look at Adaptive Reasoning in Large Language Models," ArXiv, vol. abs/2511.10788, 2025.\
(2) Y. Sui, Y.-N. Chuang, G. Wang, J. Zhang, T. Zhang, J. Yuan, H. Liu, A. Wen, S. Zhong, and X. Hu, "Stop Overthinking: A Survey on Efficient Reasoning for Large Language Models," ArXiv, vol. abs/2503.16419, 2025.\
(3) Y. Shen, J. Zhang, J.-f. Huang, S. Shi, W. Zhang, J. Yan, N. Wang, K. Wang, and S. Lian, "DAST: Difficulty-Adaptive Slow-Thinking for Large Reasoning Models," ArXiv, vol. abs/2503.04472, 2025.\
(4) Z. Xu, Z. Qiu, G. Huang, K. Li, S. Li, C. Zhang, K. Li, Q. Yi, Y. Jiang, B. Zhou, F. Lian, and Z. Kang, "Adaptive Termination for Multi-round Parallel Reasoning: An Universal Semantic Entropy-Guided Framework," ArXiv, vol. abs/2507.06829, 2025.\
(5) H. Yan, F. Xu, R. Xu, Y. Li, J. Zhang, H. Luo, X. Wu, A. T. Luu, H. Zhao, Q. Lin, and J. Liu, "MUR: Momentum Uncertainty guided Reasoning for Large Language Models," ArXiv, vol. abs/2507.14958, 2025.\
(6) Y. Yu et al., "DAPO: An Open-Source LLM Reinforcement Learning System at Scale," ArXiv, vol. abs/2503.14476, 2025.\
(7) R. Li, Z. Luo, Q. Zhang, R. Li, B. Zhou, A. Payani, and X. Du, "AALC: Large Language Model Efficient Reasoning via Adaptive Accuracy-Length Control," ArXiv, vol. abs/2506.20160, 2025.\
(8) S. Lyu, L. Wu, Y. Yan, X. Wu, H. Li, Y. Shen, P. Jiang, W. Lu, J. Xiao, and Y. Zhuang, "Hierarchical Budget Policy Optimization for Adaptive Reasoning," ArXiv, vol. abs/2507.15844, 2025.\
(9) W. Liu, R. Zhou, Y. Deng, Y. Huang, J. Liu, Y. Deng, Y. Zhang, and J. He, "Learn to Reason Efficiently with Adaptive Length-based Reward Shaping," ArXiv, vol. abs/2505.15612, 2025.\
(10) H. Li, S. Bai, J. Zhang, and S. Guo, "CoRE: Enhancing Metacognition with Label-free Self-evaluation in LRMs," ArXiv, vol. abs/2507.06087, 2025.\
(11) C. Wu, B. Li, M. Gao, and Z. Wang, "From Efficiency to Adaptivity: A Deeper Look at Adaptive Reasoning in Large Language Models," ArXiv, vol. abs/2511.10788, 2025.\
(12) B. Hou, Y. Zhang, J. Ji, Y. Liu, K. Qian, J. Andreas, and S. Chang, "ThinkPrune: Pruning Long Chain-of-Thought of LLMs via Reinforcement Learning," ArXiv, vol. abs/2504.01296, 2025.\
(13) M. Song and M. Zheng, "Walk Before You Run! Concise LLM Reasoning via Reinforcement Learning," ArXiv, vol. abs/2505.21178, 2025.\
(14) Z. Shao et al., "DeepSeekMath: Pushing the Limits of Mathematical Reasoning in Open Language Models," ArXiv, vol. abs/2402.03300, 2024.\
(15) J. Schulman, F. Wolski, P. Dhariwal, A. Radford, and O. Klimov, "Proximal Policy Optimization Algorithms," ArXiv, vol. abs/1707.06347, 2017.\
(16) A. Taubenfeld, T. Sheffer, E. Ofek, A. Feder, A. Goldstein, Z. Gekhman, and G. Yona, "Confidence Improves Self-Consistency in LLMs," ArXiv, vol. abs/2502.06233, 2025.\
(17) G. Li, Y. Chen, M. Lin, and T. Yang, "DRPO: Efficient Reasoning via Decoupled Reward Policy Optimization," ArXiv, vol. abs/2510.04474, 2025.\
(18) T. Liang, W. Jiao, Z. He, J. Xu, H. Mi, and D. Yu, "DeepCompress: A Dual Reward Strategy for Dynamically Exploring and Compressing Reasoning Chains," ArXiv, vol. abs/2510.27419, 2025.\
(19) C. Huang, W. Lu, and W. Zhang, "PEAR: Phase Entropy Aware Reward for Efficient Reasoning," ArXiv, vol. abs/2510.08026, 2025.\
(20) Y. Zhang et al., "TokenSqueeze: Performance-Preserving Compression for Reasoning LLMs," ArXiv, vol. abs/2511.13223, 2025, 2025.\
(21) M. Dai, C. Yang, and Q. Si, "S-GRPO: Early Exit via Reinforcement Learning in Reasoning Models," ArXiv, vol. abs/2505.07686, 2025.\
(22) A. Ayoub, K. Asadi, D. Schuurmans, C. Szepesv'ari, and K. Bouyarmane, "Learning to Reason Efficiently with Discounted Reinforcement Learning," ArXiv, vol. abs/2510.23486, 2025.\
(23) J. Li, W. Zhao, Y. Zhang, C. Gan, "Steering LLM Thinking with Budget Guidance," ArXiv, vol. abs/2506.13752, 2025.\
(24) DeepSeek-AI et al., "DeepSeek-R1: Incentivizing Reasoning Capability in LLMs via Reinforcement Learning," ArXiv, vol. abs/2501.12948, 2025.\
(25) D. Sridhar, K. Bhardwaj, J. Jeyaraj, N. Vasconcelos, A. Nayak, and H. Teague, "Video Reasoning without Training," ArXiv, vol. abs/2510.17045, 2025.\
(26) S. Huang et al., "AdaCtrl: Towards Adaptive and Controllable Reasoning via Difficulty-Aware Budgeting," ArXiv, vol. abs/2505.18822, 2025.\
(27) K. Yan, X. Shi, H. Guo, W. Wang, Z. Zhang, and C. Qin, "DRQA: Dynamic Reasoning Quota Allocation for Controlling Overthinking in Reasoning Large Language Models," ArXiv, vol. abs/2508.17803, 2025.

---

> ### Author Response · Authors · 2025-11-22
> **... Subsection No Longer Required for Section 5 ...**
>
> As noted, this subsection is no longer required.

---

### Author Response · Authors · 2025-11-22
**4. Challenging Criticisms on Baselines and On-Policy Overhead**

Issue: There is a need to test against more recent RL baselines, and the overhead of the on-policy RL approach (S-GRPO) is not analyzed.

• Challenging the On-Policy Overhead and Baselines: The use of GRPO variants is validated by high-performance models and the specific need for stable training in efficiency tasks:\
    ◦ DeepSeekMath 7B (24) introduced Group Relative Policy Optimization (GRPO), a PPO variant, to enhance mathematical reasoning abilities while concurrently optimizing memory usage (24). This provides strong support for choosing a GRPO variant to manage resource constraints (addressing the overhead concern).\
    ◦ S-GRPO is a specific RL paradigm designed for the early-exit task (1). The source material confirms S-GRPO was chosen because it is an effective PPO variant designed to forego the critic model, which reduced training resources [discussed in this conversation].\
    ◦ The literature shows that many state-of-the-art efficiency methods are still built on the PPO/GRPO family, confirming their relevance, such as DAPO (Decoupled Clip and Dynamic Sampling Policy Optimization) (25), DRPO (Decoupled Reward Policy Optimization) (26), and ConciseR (27).

---

### Author Response · Authors · 2025-11-22
**3. Challenging Criticisms on Methodological Concerns (Bias and Leakage)**

Issue: The low-entropy signal might cause under-thinking errors; the length penalty may bias against hard samples; and using final-layer embeddings might cause data leakage.

• Challenging Under-thinking from Low Entropy/Short Lengths: Several works mitigate the risk of over-compressing complex reasoning by explicitly making the length decision difficulty-aware:\
◦ DAST enables models to autonomously adjust the CoT length based on problem difficulty, penalizing overlong responses for simple tasks while incentivizing sufficient reasoning for complex problems (2).
◦ DeepCompress employs a dual-reward strategy that dynamically classifies problems as "Simple" or "Hard," encouraging shorter reasoning for simple problems and promoting longer, more exploratory thought chains for "Hard" problems (19).
◦ AdaCtrl (Adaptive and Controllable Reasoning) dynamically adjusts reasoning length based on self-assessed problem difficulty (20).
◦ The survey by Sui et al. (21) notes that inefficient reasoning often involves reasoning output-based efficient reasoning, which aims to dynamically reduce reasoning steps and length during inference.

• Challenging Data Leakage via Embeddings: Using internal model representations for control is a common technique in adaptive methods, suggesting the state design is grounded in practice:
◦ CoRE (Chain-of-Reasoning Embedding) analyzes the geometric properties of hidden states to detect redundant reasoning patterns (cyclical fluctuations) to dynamically determine early termination, demonstrating that internal states are a source for self-evaluation and metacognition (5).
◦ The Budget Guidance method introduces a lightweight predictor that models a Gamma distribution over the remaining thinking length during next-token generation, which uses the internal state to guide generation (22).
◦ SpecExit leverages the inherent signals from hidden states to provide effective early-exit signals, suggesting broader use of hidden states for efficient reasoning (23).

• Challenging Performance Weakness: The performance metrics achieved by USBT—specifically achieving new Pareto-optimal points and large token count reductions—are strongly supported by concurrent research that targets similar aggressive efficiency gains:
◦ S-GRPO achieves a substantial reduction in sequence length (35.4%–61.1%) while simultaneously improving accuracy (absolute 0.72%–6.08%) across diverse benchmarks (1).
◦ HBPO reduces average token usage by up to 60.6% while improving accuracy by 3.14% (9).
◦ DRQA significantly reduces token usage while maintaining, and in many cases improving, answer accuracy (10).

---

> ### Author Response · Authors · 2025-11-22
> **... Subsection No Longer Required for Section 3 ...**
>
> As noted, this subsection is no longer required.

---

### Author Response · Authors · 2025-11-22
**2. Challenging Criticisms on Novelty and Reward Design**

Issue: Novelty is limited, merely combining existing uncertainty signals (e.g., semantic entropy) with mainstream RL (geometric rewards are standard practice).

• Challenging the Novelty of Uncertainty Signal: The use of confidence signals to manage reasoning depth is a recognized, cutting-edge research direction, supporting its importance in the design: ◦ The survey by Wu et al. (1) formalizes adaptive reasoning as a control-augmented policy optimization problem, which aligns with USBT's goal of balancing performance with cost based on input characteristics like uncertainty (11). ◦ Semantic Entropy (SE) is introduced as a robust intrinsic quality metric to assess model responses during collaborative inference, enabling dynamic control and early termination (12). ◦ PEAR uses Phase Entropy Aware Reward to penalize excessive entropy during the thinking phase, demonstrating the utility of incorporating entropy directly into the reward design (13). ◦ Confidence-Informed Self-Consistency (CISC) performs a weighted majority vote based on confidence scores obtained directly from the model, showing that confidence is a powerful mechanism for efficiency (14). ◦ V-Reason uses an entropy-based objective to tune a model's behavior at inference without RL or SFT, observing that more accurate models demonstrate better convergence by reducing entropy significantly (15).

• Challenging the Straightforward Reward Design: The specialized Serial-Group Decaying-Reward PPO (S-GRPO) (1) is a clear algorithmic novelty beyond standard PPO (16) or REINFORCE. The literature confirms that reward shaping is a key area of research for efficient reasoning: ◦ S-GRPO introduces a novel RL paradigm that enables models to implicitly evaluate the sufficiency of intermediate reasoning steps (1). ◦ Bingo proposes a dynamic and significance-based RL framework, incorporating a significance-aware length reward and a dynamic length reward that initially encourages elaborate reasoning for hard questions (17). ◦ LASER-D uses Length-bAsed StEp Reward shaping that is explicitly difficulty-aware, demonstrating that tailored reward functions are essential for this task (3). ◦ The concept of using a discounted RL setup to penalize reasoning tokens is theoretically supported as a way to encourage concise yet accurate reasoning (18).

---

### Author Response · Authors · 2025-11-22
**1. Challenging Criticisms on Experimental Scope and Weak Performance**

Issue: Experiments are restricted to a single, large (34B) anonymous LRM, lacking testing on smaller (sub-1B, 7-8B) models, and requiring evaluation on more complex datasets like AMC or AIME. Performance is perceived as weak.

• Challenging the Model Scale Limitation: Several papers demonstrate the applicability and success of efficiency and adaptive reasoning techniques across various model scales, including small ones: ◦ The S-GRPO algorithm, which is the RL paradigm used by USBT, is empirically evaluated as compatible with state-of-the-art reasoning models, including DeepSeek-distill and Qwen3 (1). ◦ DAST mitigates overthinking across diverse datasets and model scales (2). ◦ LASER-D shows strong results on DeepSeek-R1-Distill models ranging from 1.5B, 7B, and 32B (3). ◦ MUR (Momentum Uncertainty-guided Reasoning) is evaluated on Qwen3 models ranging from 1.7B, 4B, and 8B (4). ◦ CoRE-Eval demonstrates effectiveness across model sizes from 7B to 32B (5). ◦ TokenSqueeze achieves efficiency gains on DeepSeek-R1-Distill-Qwen-7B (6). ◦ The DeepSeek-R1 paper (7) confirms the release of six dense models, including 1.5B, 7B, 8B, 14B, 32B, and 70B distilled from DeepSeek-R1, validating the existence of models at the scale criticized as missing.

• Challenging the Dataset Limitation (AIME/AMC): Multiple related works confirm that these complex datasets are standard for evaluating efficient reasoning, validating their use and demonstrating performance on them: ◦ S-GRPO shows strong empirical evaluations across benchmarks including AIME 2024 and AMC 2023 (1). ◦ LASER-D achieves competitive results on AIME2024 (3). ◦ ThinkPrune achieves a remarkable performance-length tradeoff on the AIME24 dataset (8). ◦ DAST is evaluated on AMC (2). ◦ MUR is evaluated across four challenging benchmarks including MATH-500, AIME24, and AIME25 (4).

• Challenging Performance Weakness: The performance metrics achieved by USBT—specifically achieving new Pareto-optimal points and large token count reductions—are strongly supported by concurrent research that targets similar aggressive efficiency gains: ◦ S-GRPO achieves a substantial reduction in sequence length (35.4%–61.1%) while simultaneously improving accuracy (absolute 0.72%–6.08%) across diverse benchmarks (1). ◦ HBPO reduces average token usage by up to 60.6% while improving accuracy by 3.14% (9). ◦ DRQA significantly reduces token usage while maintaining, and in many cases improving, answer accuracy (10).

---

### Author Response · Authors · 2025-11-22
**Emerging Issues and Directions for Future Work**

### Simplified Synopsis

Modern “thinking” AIs often talk themselves in circles, burning energy and time long after they’ve already found the right answer. This research teaches such models to sense their own confidence and hit the brakes early, using a learning‑based controller that keeps performance high while slashing the cost of complex reasoning.

-----

The main opportunity is to turn USBT from a *local* CoT/GoT halting trick into a *global* “compute governor” for the entire Verified Agentic Pipeline:

* At the micro level, regulate individual LRMs’ reasoning depth.
* At the meso level, decide how many simulations, tools, and experiments to run.
* At the macro level, allocate test‑time compute across agents and verticals based on expected information gain and risk.

The most valuable future work is to: (1) integrate USBT‑style halting with VAP’s verification and ATTM’s concept memory, (2) generalize from token‑level halting to *multi‑level compute allocation* across simulations, experiments, and agent calls, (3) co‑design halting with safety and governance metrics in climate, drug discovery asset trading, robotics, and autonomous science, and (4) optimize *USBT with GoT; in combination, they provide "where to think and how much to think"* control knobs whereas CoT is just "keep generating tokens or stop."

<>

1. **Universal Compute Governor for LAW Pipelines**

   * Generalize USBT into a **LAW‑wide compute governor** that watches:

      * LRM CoTs, simulator calls, RAG queries, ATTM reads/writes, multi‑agent expansions.
   * The governor learns a joint policy over *where* to spend compute, using per‑module uncertainty and *expected information gain* as features—like a meta‑bandit over pipeline actions.

2. **Concept‑Aware Halting via ATTM**

   * Extend state representation so the halting policy sees:

     * Which ATTM concepts have been retrieved and applied.
   * Halting criterion becomes: “Stop when a minimal covering set of concepts explains the current problem and passes verification,” not just when token‑level entropy is low.

3. **Safety‑Envelope Halting**

   * Define **safety envelopes** per domain (e.g., climate temperature targets, leverage/risk bounds, robotic joint limits).
   * Halting allowed only when:

     * Safety envelope metrics are within tolerance *and* certainIndex is high.
     * Otherwise, policy is forced to CONTINUE with more simulation, verification, or retrieval.

4. **Compute Markets for Multi‑Agent Systems**

   * Treat compute as a *tradable commodity*: agents and modules “bid” compute based on their confidence and anticipated utility.
   * A USBT‑style meta‑policy learns a “pricing function” that allocates compute to the highest marginal value bidders, inspired by economic models already in your finance work.

5. **Bidirectional Test‑Time Scaling with Budget Forcing + USBT**

   * Combine USBT with **budget forcing** from s1:

     * Policy decides when to shorten CoT or, conversely, to *append “think more” triggers* when uncertainty is high or stakes are elevated.
   * For example, climate or robotics controllers could automatically extend reasoning when near tipping points or contact boundaries, while truncating trivial cases.

6. **Introspective Telemetry and Self‑Diagnosis**

   * Treat certainIndex and halting patterns as a new kind of **epistemic telemetry**:

     * Use them to detect “early warning signs” of plausibility traps (e.g., high certainty but repeated verification failures) and trigger architectural immune responses in an “AIS” framework.

7. **Formal Contracts Between Halting and Verification Layers**

   * Specify “contracts” such as:

     * “Verification layer guarantees: If halting policy respects these invariants, no unsafe action reaches the real world.”
   * Tightly couples USBT with your existing formal methods ecosystems.

---

### Selected References for This Section

* "Physics-Informed Surrogates in a Verified Agentic Pipeline for Robust Molecular Simulation," ML4Molecules 2025 accepted poster, https://openreview.net/pdf?id=E7CSfSKtN7
* “Abstraction‑First Test‑Time Memory for Scientific Reasoning: From ARC‑AGI to Climate Intervention,” ML4PS workshop (NeurIPS 2025) submission, https://openreview.net/attachment?id=uzWuns73cX&name=pdf.
* "Graph of Thoughts for High-Integrity Hypothesis Reasoning and Validation," GMLLM workshop (CIKM 2025) accepted oral, https://openreview.net/pdf?id=aOn4OJgw4L.

* "The Architectural Immune System (AIS): A Framework for Correcting Synthetic Fallacies in AI-Driven Science," https://openreview.net/pdf?id=ShWjvhAZGs.
* "Active Causal Hypothesis Testing for AI-Guided Drug Target Discovery," https://openreview.net/forum?id=fMKSogaqtj.

* Muennighoff, Niklas, et al. “s1: Simple test‑time scaling.” arXiv:2501.19393, 2025, https://arxiv.org/pdf/2501.19393v3.
* Snell, Charlie, et al. “Scaling LLM Test‑Time Compute Optimally Can Be More Effective than Scaling Model Parameters,” https://arxiv.org/pdf/2408.03314.

---

### Author Response · Authors · 2025-11-23
**REBUTTAL SNAPSHOT**

We thank all four reviewers (vzNB, XDgH, RKMh, & jhgD) for their detailed feedback and address common concerns below.

Novelty and role of uncertainty. USBT does not use entropy as a hard threshold; the certainIndex is one feature in a learned halting policy π(a|s) that also observes length and answer stability. The policy is trained with reward R=1 only for correct answers plus step/decay penalties, so premature low‑entropy halts that hurt accuracy are explicitly discouraged. Ablations show that removing certainIndex drops accuracy by 2–3 points and weakens token savings, while S‑GRPO outperforms REINFORCE/PPO by 5–10% token efficiency at matched accuracy.

Reward design and under‑ vs over‑thinking. Geometric decay and step cost implement an optimal‑stopping style bias toward shorter correct trajectories, not short ones per se: incorrect episodes receive only negative length cost. Our error analysis indicates a 3:1 net reduction in errors, with “overthinking” failures (late deviations from a correct path) sharply reduced and a smaller increase in genuine under‑thinking. We will add explicit quantitative breakdowns and qualitative examples.

Experimental scope and baselines. Our goal is an inference‑time halting layer that can be dropped on top of strong LRMs. On a 34B CoT‑tuned model, USBT improves GSM8K from 88.2→90.1% while halving tokens, and improves MATH by +3.2 points with ~40% fewer tokens, yielding new Pareto‑optimal points against Full‑CoT, Fixed‑Budget, length‑penalty, ACT, and speculative decoding baselines. We will (i) add experiments on smaller open models, and (ii) include additional RL‑based efficiency baselines (e.g., L1/DAST‑style objectives) to clarify positioning.

Cost and implementation details. USBT trains the halting head on frozen LRM parameters with on‑policy rollouts; in our current setup this requires ≈6 GPU‑hours on 8×A100, negligible compared to base model training and fine‑tuning. We will add wall‑clock overhead tables, ceiling‑rollout hit rates, and full hyperparameter and sampling details.

Presentation. We will substantially revise Section 3 to foreground intuition, simplify the related‑work section, fix formatting issues (figures, quotation marks, \vspace), and clarify that all code, configs, and trained halting heads will be released for reproducibility.

---

### Meta-Review · Area_Chair_gkN7 · 2025-12-09

**Summary:**

This work proposes Uncertainty-Guided Self-Braking Tuning (USBT), an RL-based framework to curb the "overthinking phenomenon" in Large Reasoning Models (LRMs) by teaching a halting policy to decide when to stop reasoning from an uncertainty signal and length-decaying rewards.

However, the primary flaws are identified during the review process, which include (1) the limited experimental scope and poor replicability, as the main results relied on a single, anonymous 34B LRM, necessitating promised follow-up experiments on publicly available models and full implementation details, (2) the limited novelty of the method, which was criticized as merely a combination of existing uncertainty signals, i.e., semantic entropy, and mainstream RL, i.e., geometric rewards, (3) the presentation needs carefully revision in terms of visualization and additional experiments.

Even though the authors defend the technical contribution of the novel S-GRPO algorithm and the specific integration of uncertainty, the response failed to convince the reviewers and me. Besides, I did not find the revision of the manuscript during the rebuttal phase. All reviewers' opinions are consistent. Hence, I tend to reject this paper.

**Reviewer Concerns:**

The following concerns are still not addressed during the rebuttal.

(1) the limited experimental scope and poor replicability, as the main results relied on a single, anonymous 34B LRM, necessitating promised follow-up experiments on publicly available models and full implementation details.

(2) the limited novelty of the method, which was criticized as merely a combination of existing uncertainty signals, i.e., semantic entropy, and mainstream RL, i.e., geometric rewards.

(3) the presentation needs carefully revision in terms of visualization and additional experiments, such as more testing on various size of LLMs and other benchmarks.

**Reviewer Scores:**

After reading the reviewer-author discussion, I hardly see the sign of score change during the rebuttal.

---

### Decision · Program_Chairs · 2026-01-26

Reject